# Help Needs among Parents and Families in Times of the COVID-19 Pandemic Lockdown in Germany

**DOI:** 10.3390/ijerph192114159

**Published:** 2022-10-29

**Authors:** Christiane Baldus, Simone Franz, Rainer Thomasius

**Affiliations:** German Centre for Addiction Research in Childhood and Adolescence, University Medical Centre Hamburg-Eppendorf, Martinistraße 52, 20246 Hamburg, Germany

**Keywords:** COVID-19, pandemic, family functioning, family help need, parent mental health, pandemic-related stress, child health-related quality of life

## Abstract

Background: The COVID-19 pandemic was accompanied by multiple disruptions in the everyday lives of families. Previous research has underlined the negative impact of the pandemic on stress among parents and identified factors related to heightened levels of stress. Yet, several potential stressors have not been taken into account. Moreover, little is known about how general and pandemic-related stressors impacted help-seeking intentions for personal or family problems. Methods: We recruited N = 602 parents and their children (*n* = 101) for a cross-sectional online survey on parent, child and family well-being, stress and help need after the first wave of COVID-19 infections in Germany. Data were analysed using multinomial regression analyses to predict family help need, taking into account pre-pandemic help-seeking. Results: Parents showed high levels of stress, which were associated with pre-pandemic mental health, family functioning, pandemic related worries about finances, household workload and health worries. While 76.2% of families reported no during-pandemic help need, 11.3% reported a help need before and during the pandemic and 12.5% of families without prior help needs reported a new help need during the pandemic. Conclusions: The results of the present study underline the need for help service providers to adapt their offers.

## 1. Introduction

The emergence of the novel SARS-CoV-2 virus and the COVID-19 pandemic [1] were associated not only with a global health threat but also with a profound disruption of everyday life. In Germany, a rise in infections in March 2020 [2] led the government to close down large parts of public life. This fairly unprecedented situation posed a challenge, especially to families. Research has shown a significant impact of the COVID-19 pandemic on family well-being [3]. Gaining knowledge about the risk factors associated with child, parent and family stress is a first important step to understand the pandemic’s impact on families. It is further imperative to support families in light of a crisis, especially those who report struggling with the situation and being in need of support. Therefore, as a second step, characterising families who report a help need at the beginning of the COVID-19 pandemic is helpful to understand families’ situations and to be able to inform service providers about how to target family needs in a pandemic situation. Drawing from a large cross-sectional sample of parents and children in the first months of the COVID-19 pandemic in Germany, the current study seeks to add insight into associations between different stressors in families and to gather information on families with a self-reported help need during the pandemic. We were particularly interested in looking at how families who reported problems before the pandemic occurred were doing, as well as those families who had no previous need for help.

The beginning of the COVID-19 pandemic was associated with a variety of concerns for people around the world [4]. Measures of virus containment relied largely on social distancing (“lockdown”), including the closure of large parts of public life, such as retail, daycare and schools. The associated economic disruptions [5,6] put many of those affected under economic pressure. Many children attended schooling from home [7] thus demanding higher levels of parental supervision. Families in Germany, who often rely on social networks, (e.g., grandparents) to handle the demands of work life and childcare [8], had to adjust to the situation while reducing contacts These conditions of uncertainty, loss of control, isolation, daily routine disruptions, added workload and economic pressure set conditions for experiencing high levels of stress [9] within families.

At the same time, there is no doubt that families play a key role in any society. Family functioning, emotional climate and parental health have a lasting impact on child development [10]. Service provision to parents and families at risk of dysfunction is therefore important to foster family and child well-being. Ample research before the COVID-19 pandemic identified risk factors for family dysfunction. Those general factors associated with family dysfunction include high family conflict [11], poor parenting practices [12,13] and poor parent–child relationships. Deprived families, for example those living in poor neighborhoods [14] or experiencing economic pressure [15], are often among those with heightened levels of dysfunction. Yet, economic disadvantage itself does not seem to impact child development rather its combination with parental mental health problems, including substance use problems [16,17,18], interparental conflict and parent stress [19]. Specifically, parental stress associated with economic pressure is known to increase parental mental health problems and interparental conflict. It is then interparental conflict which strains parent–child attachment security [20] and parenting styles [15,19] resulting in poorer child adjustment [21].

Several of these stressors, which are known to negatively impact family life, are assumed to have exacerbated during the COVID-19 pandemic, such as economic pressure, parental stress and mental health problems. In fact, data on the impact of the COVID-19 pandemic show significant decreases in health-related quality of life, satisfaction with family life and family well-being [22,23] and increases in parental mental health symptoms [22]. There were also reports of lower levels of positive parenting experiences [3,24]. Regarding child well-being, there is some mixed evidence regarding changes in the earlier months of the pandemic; while some studies found increased levels of internalizing and externalizing problems in children [22], others found no difference in psychosocial adjustment during the pandemic as compared to norms [3].

Several risk factors have been identified to be associated with higher levels of adjustment problems in parents and families in the COVID-19 pandemic. Among them are single parents [3], families with a higher number of underage children [3], families with changed working arrangements [3,24,25] and parents with mental health problems prior to the pandemic [24,25]. Mothers reporting financial strains were found to show elevated risks of depression [24,26]. At the same time, maternal depression was linked to problems balancing working and homeschooling requirements and difficulties in obtaining childcare [26]. Marital quality was associated with lower anxiety and depression among mothers [24]. A deterioration of parent–child relationships during the pandemic was associated with lower levels of positive parenting experiences [3]. Parents with lower levels of educational attainment reported a larger decrease in family life satisfaction [23] and higher levels of depression [24]. Other results, however, found the relative increase in depression among parents was larger in families commonly regarded as low-risk, that is those with higher education levels and low levels of coparenting conflict [22].

Families living in challenging situations may face several barriers to seeking informal or formal help. Barriers are known to include fear of social stigma [27,28], lack of knowledge about where to obtain help [29], preference of handling challenging situations alone [30,31], difficulties recognizing problems [30] or a hope that problems might dissolve [31]. A greater proclivity to seek help was found in parents with higher education and income [32]. Help seeking, however, is influenced by the context, in which a help need arises; hesitancy to access help is lower in emergency situations, such as in light of natural disasters [33].

The COVID-19 pandemic can be regarded as a natural disaster. The limited literature on family help seeking during the COVID-19 pandemic so far shows that parent help seeking was higher for those reporting higher levels of mental health symptoms such as depression [34] or anxiety [35]. For mothers with heightened levels of anxiety or depression, barriers to contacting mental health services during the COVID-19 pandemic include a lack of time and energy, cost of services, no recognised need for services, uncertainty as to how to contact services, lack of interest and doubts services could help [24].

Generally, knowledge about parent stress, family functioning during the pandemic and associated stressors needs to be further expanded in order to better understand the challenges families face during the COVID-19 pandemic. Moreover, we are interested in characterizing families with a self-reported, during-pandemic help need for family or personal problems, because this may inform targeted interventions. At the same time, irrespective of the emergence of any public health crisis, there are families who struggle with problems and are in need of support. We assume that families with a history of dysfunction and a previous need for help may be hit especially hard by the pandemic. Among the families with a help need, we expect to see families who had also reported help needs before the COVID-19 pandemic. At the same time, and with this new crisis evolving, it is important to find out if there are families who had not accessed previous help but now—in the face of the crisis—identify as struggling and report a new help need. Within the framework of the current study, we seek to be sensitive to whether families had already reported a previous help need. By adjusting our approach to be sensitive to the possible presence of pre-pandemic help needs, this paper adds a new aspect to the knowledge about families who struggle during the pandemic. We assume three groups of families can be identified with respect to pre- and during-pandemic help need: (a) those families who do not report any during-pandemic help need, (b) those who report a pre- and during-pandemic help need (continuous help need) and (c) families with a new help need in the pandemic (no pre-pandemic help but during-pandemic help need). With this historically unprecedented pandemic evolving, we are interested in exploring the interplay between, and the relative impact of, known general risk factors for family dysfunction and pandemic-specific factors. Possibly there are specific pandemic-related stressors which play a central role, such as the surge in working from home, the need to balance work responsibilities with caring and homeschooling for children or aggravated health worries. While we assume that families with a continuous help need report known risks to family dysfunction, we are interested to find out which pandemic-related stressors play an additional role for these families. For families with a new, during-pandemic help need, we want to find out which factors contribute to their help need. Are these previously seemingly well-adjusted families specifically strained by pandemic-related stressors, or are there other general risk factors for family dysfunction playing a role as well?

We hypothesise that among families with continuous help needs, there is a heightened proportion of families, who are generally known to report high stress levels, specifically reporting pre-pandemic mental health problems, substance use problems, being single parents and having low income. We also seek to explore the role these factors possibly play in families with a new help need. We further hypothesise that families with a during-pandemic help need show elevated scores on variables that have previously been associated with high family stress in the pandemic. Those variables include previous mental health problems, families with younger children, single parents, those with disruptions in work, financial strain and those reporting lower satisfaction with partner or family relations. We therefore expect parents with a during-pandemic help need to report higher levels of during-pandemic stress and during-pandemic mental health symptoms, higher levels of pre-pandemic mental health problems, heightened financial strain, work disruptions, single-parent status, strained family and partner relations and having younger children. Moreover, we are interested in the impact of further pandemic-associated stressors on family and parent stress, namely strain from working from home, household workload, health worries, restricted housing and relationship changes with friends.

The aim of this article is to gain insight into which factors are associated with parental stress and to characterise families who report a help need during the first six month of the COVID-19 pandemic in light of possible previous help needs, and to draw comparisons to families with no reported help need. Our principal research question is: What contributes to families’ perceived need for help during the COVID-19 pandemic? To achieve this, we seek to (1) explore risk factors and pandemic-related challenges which are associated with higher levels of during-pandemic parental stress and strain, (2) explore how many families there are who report a new during-pandemic help need while having not reported a pre-pandemic help need (new help need), (3) compare the families who report a new help need during the pandemic to families who report no help need, or to those who report a continuous help need, that is a help need before and during the pandemic. Moreover, we want to (4) identify sets of risk variables which predict membership of either group of during-pandemic help need (either new or continuous). In addition, we seek to find out whether pandemic-related stressors help to identify families with a during-pandemic help need when adjusting for predictors which are generally known to be associated with family dysfunction.

## 2. Materials and Methods

### 2.1. Study Design and Procedure

The study CovFam was conducted in Germany as a nationwide cross-sectional online survey between 5 June and 26 July 2020. After recruitment, data of N = 602 parents and their children (*n* = 101) were available. The study was approved by the Chamber of Physician’s Ethics Committee. All families with minor children living in Germany were eligible to participate. Families were contacted via youth and family support organizations, family and parent interest groups, social and youth welfare organizations, churches, employers and private contacts. Supporting institutions or persons distributed study information and the invitation link, e.g., via newsletters, social media posts or mailing lists, each with the invitation to share study information further in own networks. On the study’s landing page, families were informed about the study and informed consent was obtained. Parents were asked anonymously about their experiences and their children’s well-being. Thereafter self-reports of children were gathered from children and adolescents aged between 11 and 17 years. In a pilot run, parents and children took about 40 min each to complete the survey.

### 2.2. Measures

#### 2.2.1. Sociodemographics

We collected data about educational background, family composition with regards to number and respective age of children, housing (square meters/person) and single-parent status. Families were also asked to give their household’s mean monthly income up until February 2020 and whether their income changed in April 2020, and to give an estimate of income loss (in %) if applicable.

#### 2.2.2. Help-Seeking Behaviour

Help seeking during the pandemic. We also asked whether families had considered seeking help for family or personal problems in April 2020 (“Have you sought help for family or personal worries or strains in April 2020?” (No, I had no need./Yes, I wanted to seek help, but it was not possible due to the pandemic./Yes, I sought out help through personal contact./Yes, I sought out help through technically supported contact (by telephone or online)).

Pre-pandemic help seeking. We assessed family help-seeking behaviour prior to the pandemic by asking “Have you used services for help, (e.g., counselling, psychological services) before March 2020?” (Yes/No).

#### 2.2.3. Parental Health (Pre- and during-Pandemic)

During-pandemic parent mental health. Parental mental health problems were measured using the SCL-K-9 [36], a 9-item short version of the Symptom Checklist-90-R by Derogatis (German version: [37]). The 9 items comprise symptoms of somatization, obsessive-compulsive experiencing, interpersonal sensitivity, depression, anxiety, anger/hostility, phobic anxiety, paranoid ideation and psychoticism, and are rated on a 5-point Likert scale (0 “not at all” to 4 “very severe”). The mean across all 9 domains can be interpreted as a measure for general mental health problems, with a higher score indicating more problems. Pre-pandemic help seeking. We assessed family help-seeking behaviour prior to the pandemic by asking “Have you used services for help, (e.g., counselling, psychological services) before March 2020?” (Yes/No).

During-pandemic parent’s perceived stress. Perceived stress was measured with the Perceived Stress Scale [38], a German version of the original PSS-10 by Cohen and colleagues [39]. Both parents and children indicated their feelings of control, helplessness and general stress on 10 items using a 5-point Likert scale with scores ranging from 0 “never” to 4 “very often”. A sum score can be calculated; higher scores point to higher levels of perceived stress.

Parent’s previous mental health and substance use issues. We used several variables, which were originally devised by the CRISIS group and adapted them to suit the present study. Among those questions are items measuring parent mental health problems (“Has a medical/psychiatric/psychological professional ever related to you that you had a mental health disorder?” Yes/No), parent substance use problems (“Has a medical/psychiatric/psychological professional ever related to you that you had an alcohol/illicit drug use problem?” Yes/No).

Parent’s self-reported alcohol use. We assessed parent alcohol use using the AUDIT-C, a 3-item screening tool for potentially hazardous alcohol use [40]. The AUDIT-C focuses on three aspects of alcohol use: frequency of alcohol use (0 “never”, 1 “monthly or less”, 2 “2 to 4 times a month”, 3 “2 to 3 times a week”, 4 “4 times a week or more”), quantity of alcohol use and frequency of binge drinking, defined as drinking 6 or more alcoholic beverages on one occasion. Different cut-offs are used to identify individuals with a hazardous drinking pattern [41,42]. For the current paper we followed the approach by Varnaccia and colleagues [43] and used a score of 4 or higher for females and 5 or higher for males to identify at-risk drinkers.

Change in alcohol use was assessed by asking, “Has the amount and frequency of your alcohol use now changed compared to before March 2020?” (1 “much less”, 2 “somewhat less”, 3 “unchanged”, 4 “somewhat more” and 5 “much more”). The same question was used in a similar fashion by other authors [44].

Change in parent physical and mental health was assessed by asking, “How would you rate your overall physical health up until February 2020?” and “How would you rate your overall mental/emotional health up until February 2020?” (1 “excellent” to 5 “poor”). Both questions were asked with respect to current health status with the same answer format. The questions are a slight modification of questions developed by the CoRonavIruS Health Impact Survey (CRISIS) V0.2 (www.crisissurvey.org, accessed on 4 May 2020) [45]. We used a change score by subtracting scores from each other with negative scores indicating a decline in health and positive scores indicating improved health for both somatic and mental health.

#### 2.2.4. Family Relationships

Strain in partner/child relationship. We asked how much strain participants felt due to pandemic-related change regarding their relationship with their partner and children in two separate questions: “… how stressful have these changes in contacts with your partner been for you?” and “… how stressful have these changes in contacts with your child(ren) been for you?” (1 “not at all” to 5 “extremely”). Both questions are modelled after questions developed by the CoRonavIruS Health Impact Survey (CRISIS) V0.2 [45].

Family functioning. The family APGAR is a widely used, validated and reliable 5-item measure to assess family functioning [46,47]. Questions are assessed with regards to family adaptation, partnership, growth, affection and resolve on a 3-point Likert scale (1 “not at all” to 3 “always or nearly always”). A high sum score across the 5 items indicates good family functioning. For the current analyses we used the parents’ answers on family functioning.

#### 2.2.5. Pandemic-Related Stressors

Pandemic-related stressors were assessed by asking, “Since the COVID-19 pandemic has changed our everyday life, stress and strain from “… worries about finances”, “… working from home”, “… household workload”, “… health worries (own health and that of significant others)” have changed.” Answers were given on a 5-point Likert scale ranging from 1 “much less strain” to 5 “much more strain”.

Strain in relationships with friends. We asked how much strain participants felt due to pandemic-related change regarding their relationships with their friends; “… how stressful have these changes in contacts with your friends been for you?” (1 “not at all” to 5 “extremely”). The question is modelled after a question developed by the CoRonavIruS Health Impact Survey (CRISIS) V0.2 [45].

Time spent outside the family home. Participants were asked how much time they had spent outside the family home on average on a regular weekday in April 2020, in hours.

#### 2.2.6. Child Self-Reported Well-Being

Child during-pandemic health-related quality of life was assessed using the self-report version of the KIDSCREEN10, a 10-item measure assessing children’s quality of life with regards to mental and physical health, family, friends, leisure time and school. Children indicate their quality of life on a 5-point Likert scale ranging from 1 (“not at all/never”) to 5 (“extremely/always”). A total quality of life index can be calculated. German norms are available [48].

Child mental health was assessed using the German self-report version of the Strengths and Difficulties Questionnaire [49]. The 25 items of the SDQ allow for measuring 5 domains of psychosocial functioning: emotional problems, conduct problems, peer problems, hyperactivity/attention problems and prosocial behaviour in the past 6 months. The first 4 scales can be combined into an aggregated total problem scale. Each of the 25 items is assessed on a 3-point scale ranging from 0 (“not at all”) to 2 (“certainly true”). Higher scores point to higher levels of problems or prosocial behaviour. German norms allow for a classification of total problem scores into a normal, borderline or abnormal category [49].

### 2.3. Analytic Strategy

All analyses were conducted using IBM SPSS Statistics Version 27. Little’s Missing Completely at Random (MCAR) test indicated that variables regarding pre-pandemic mental health, during-pandemic mental health, pandemic-related stressors and help need were completely missing at random. Significance was indicated by *p* < 0.05 for all statistical tests. In a first step, we compared our sample data to other representative family studies from Germany to gain insight into possible sampling bias and into norms regarding during-pandemic stress levels. Moreover, we sought to find out how many families reported previous and during-pandemic help need. In a second step, we used Pearson and point biserial correlations to identify relationships between variables of stress, during-pandemic mental health problems and known family risk-factors and pandemic-related stressors.

In a next step, we used ANOVAs, Tukey post-hoc tests, Kruskal-Wallis-tests and z tests to compare different help-seeking family groups (no help need/continuous help need/new help need) with regards to family characteristics and family and pandemic-related stressors. Multinomial logistic regressions were then conducted to predict families’ during-pandemic help need from variables demonstrated to be relevant in prior group comparisons using families with no during-pandemic help need for reference. In particular, we sought to find out which pandemic-related stressors are most relevant in families with a new or a continuous during-pandemic help need. We also wanted to assess what role pandemic-related stressors play in relation to known family risks, especially during-pandemic stress and pre-pandemic mental health problems. To achieve this, predictors of during-pandemic help need were entered blockwise, starting with pandemic-related stressors, then entering during-pandemic stress in a second step, pre-pandemic mental health problems and family relationships last. We believe moving from more proximal pandemic-related stressors to more overall distal stressors allowed us to find out about the relative impact of stressors. Nagelkerkes R^2^ and maximum likelihood estimations were used to assess model fit and likelihood ratios were used to assess if using added sets of predictors improved models [50].

## 3. Results

### 3.1. Participant Characteristics and Economic Impact of COVID-19

A total of N = 602 families completed the survey data. Parents were on average 43.3 years old (SD = 6.56). Answering parents were mostly mothers (85.2%). Data are available for *n* = 101 children and the child mean age was 13.3 years (SD = 2.15). Of the children, 48.5% identified as female, 1.0% as diverse and 50.5% as male. Most families reported having two children (48.2%), while 32.4% reported having one child and 19.4% having three or more children. With regards to economic pressure, 26.9% of families reported an income loss during the COVID-19 pandemic. Among those reporting income loss, the average reduction of income was 22.63% compared to before the pandemic (SD = 19.61).

### 3.2. Parent Health and Child Well-Being

Parental stress levels (PSS-10) in the present sample during the pandemic were M = 18.26 (SD = 7.11) in mothers and M = 16.89 (SD = 6.56) in fathers. With regards to alcohol use, 20.3% of mothers and 31.4% of fathers scored above the AUDIT-C’s cut-off for risky use. Participating children reported a quality of life on the KIDSCREEN10 of M = 34.52 (SD = 5.18) and a total problem score on the SDQ of M = 10.41 (SD = 4.91; Table 1).

### 3.3. Correlates of during-Pandemic Stress and Mental Health Symptoms

Associations between during-pandemic parental stress and mental health symptoms, family risks and pandemic-associated variables of interest are reported in Table 2. Parental stress was associated with single-parent status, the presence of children under the age of six, pre-pandemic mental health and substance use problems, a strained partner relationship, strained relationships with friends, family functioning, worries about finances, stress through working from home, household workload and health worries. Parental stress was not related to family housing, income loss or a strained child relationship. A rather similar pattern emerges for correlations between parental mental health symptoms. Associations were found between single-parent status, pre-pandemic mental health and substance use problems, a strained partner relationship, strained relationships with friends, family functioning, worries about finances, household workload and health worries. A small association was also found between financial worries and income loss (r = 0.151). A strained partner relationship was associated with a strained child relationship, with a medium-size effect (r = 0.419). Interestingly, strain in relationships with both a partner and child was associated with strain in relationships with friends too (r = 0.296 and r = 0.272), again with medium size effects. A strained partner relationship was also associated with worries about finances (r = 0.157) and household workload (r = 0.174), both with small effects. Household workload also correlated with strained child relationships (r = 0.130) and working from home (r = 0.259) and was further associated with lower levels of family functioning (r = −0.180). More health worries were found in single parents (r = 0.123). Health worries were also related to strained relationships with friends (r = 0.172) and family functioning (r = −0.149), but not to partner or child relationships. Health worries were further associated with worries about finances (r = 0.228) and household workload (r = 0.143).

### 3.4. Help Seeking and Help Need

Of the 602 participating families, 424 answered questions with regard to previous help seeking and during-pandemic help need. Among them, 323 reported no help need (76.2%). Of these families, 31 reported they had used previous help but reported no during-pandemic help need at the time of assessment. Among the families with a during-pandemic help need, 48 families indicated they had used help offers before the pandemic (11.3% continuous help need) and 53 families who had not reported a pre-pandemic help need do so now during the pandemic (12.5% new help need).

### 3.5. Comparison of Families with No Help Need, Continuous Help Need and New Help Need

Characteristics of families with no during-pandemic help need, with continuous help need and a new during-pandemic help need are reported in Table 3. Average during-pandemic stress levels on the PSS-10 and during-pandemic mental health symptoms on the SCL-K-9 were significantly higher in families with a during-pandemic help need. The highest scores were found among families with a continuous help need (Table 3). Families with a continuous help need were more often single parents (47.9%) and had the highest rates of pre-pandemic diagnosed mental health (56.25%) and substance use problems (14.58%). Families with a new help need also reported higher rates of pre-pandemic mental health problems (30.19%) compared to families with no help need (13.00%), but lower rates than families with a continuous help need (56.25%). Substance use problems did not play a role in families with a new help need (0.00%). Family relationships were most functional in families with no help need and did not differ from families with a new help need. However, both of these groups reported more functional family relationships than families with a continuous help need, who reported the lowest family relationship scores on the Family APGAR. Pandemic-related stressors played a different role among family groups. Family groups with a during-pandemic help need reported higher household workload and health worries compared to families with no help need. Significant decreases in levels of parent mental and physical help were obtained in all three family groups. However, they played a bigger role in families with a new help need, as did worries about finances. Families, however, did not differ significantly with regards to actual income losses. There were no differences across family groups regarding worries about job security, working from home or the proportion of families who did not leave the house at all.

### 3.6. Prediction of during-Pandemic Help Need in Light of Previous Help Need

Block-wise multinomial logistic regressions were conducted to predict during-pandemic help need groups (Table 4) in four blocks: pandemic-related stressors (block 1), followed by during-pandemic mental health (block 2), pre-pandemic mental health problems (block 3) and family relationships (block 4). In block 1 we focused on variables that had shown group differences before (Table 3). Families with no during-pandemic help need served as reference group. During-pandemic stress (PSS-10; block 2) and during-pandemic mental health (SCL-K-9; block 2) correlated highly (r = 0.729), as did during-pandemic stress (PSS-10; block 2) and change in parent mental health (block 1; r = 0.614). To avoid problems with multicollinearity we only used during-pandemic stress as a predictor. Pre-pandemic parental substance use problems were not prevalent in the new help need group and were dropped in multinomial regressions to improve model stability. Thus, a first model (model 1) included change in parent physical health, household workload, strained child relationship and worries about finances. For families with a continuous help need, health worries were the only predictor that contributed significantly to the model (OR = 2.761, 95% Confidence Interval (CI) [1.643, 4.640], *p* = 0.000). For families with a new help need the sole significant predictor was change in parent physical health (OR = 0.420, 95% CI [0.269, 0.658], *p* = 0.000). When during-pandemic stress was added (model 2), the prediction was significantly improved (Likelihood ratio = 7.285, df = 2, *p* = 0.026), and during-pandemic stress was a predictor for both families with a continuous help need (OR = 1.229, 95% CI [1.144, 1.320], *p* = 0.000) and a new help need (OR = 1.092, 95% CI [1.027, 1.160], *p* = 0.005). Moreover, continuous help need was additionally predicted by strained child relationship (OR = 1.247, 95% CI [1.013, 1.535], *p* = 0.037). Model 3 added pre-pandemic mental health problems, and again, the prediction could be significantly improved compared to model 2 (Likelihood ratio = 15.321, df = 2, *p* = 0.000). Pre-pandemic mental health problems were a predictor for both continuous help need (OR = 4.796, 95% CI [2.052, 11.209], *p* = 0.000) and new help need (OR = 2.912, 95% CI [1.239, 6.845], *p* = 0.014). Still, the other during-pandemic stressors kept their significance, except for strained child relationships (see Table 4). A further inclusion of family relationships (Family APGAR; block 4) did not improve the model (Likelihood ratio = 0.803, df = 2, *p* = 0.669).

## 4. Discussion

### 4.1. Comparison to Pre-Pandemic German Data on Parent Health and Child Well-Being

The present study adds to the growing evidence regarding the impact of COVID-19 on the well-being of families during the first wave of the COVID-19 pandemic. When comparing different aspects of psychosocial adjustment in the present sample to pre-pandemic representative samples of parents, we found significantly higher scores of PSS-10 stress levels in both mothers (M = 18.26; SD = 7.11) and fathers (M = 16.89; SD = 16.89) as compared to a previous representative German sample (women M = 13.07, men M = 12.62; [38]). These results are in line with previous results regarding well-being in the pandemic, which had shown significantly lower levels of well-being overall, and specifically among females and mothers [24,52]. Rates of during-pandemic risky alcohol use lay in the range of previous pre-pandemic studies [43] with 20.3% of mothers and 31.4% of fathers in the present sample scoring above the AUDIT-C cut-off for risky alcohol use. With regards to child well-being, present data on child psychosocial problems were comparable to pre-pandemic German SDQ data with a total problem score of M = 10.41 (SD = 4.91) in the present sample and M = 10.37 in the reference sample [49]. The only differences between the present and the reference sample in SDQ data were found in the subscale conduct problems, in which the present sample reported a lower average (M = 1.48; SD = 2.61) compared to M = 1.74 in the reference sample. The current sample, however, shows markedly lower levels of children’s self-reported quality of life on the KIDSCREEN10 during the pandemic (M = 46.70; SD = 9.21) in comparison to pre-pandemic German norms, in which a mean score of M = 49.85 was obtained [48]. Similar results regarding lower levels of quality of life in children during a similar period of time within the pandemic in Germany have been reported [53]. Unlike that study, however, significantly higher levels of child mental health problems as measured by the SDQ could not be observed in the present sample. Yet, our present result is similar to other previous research [3]; the evidence here seems to remain mixed. We believe that current findings regarding the diminished quality of life in children seem plausible given the numerous constraints on children’s daily activities during the pandemic. Among them are reduced and/or remote schooling, reduced socializing with friends and extended family and very limited options for cultural and leisure activities. On the other hand, impressions of the pandemic might have been too recent and fairly time-limited to result in heightened levels of mental health problems in children at this rather early point within the pandemic.

### 4.2. Findings on Correlates of during-Pandemic Stress and Mental Health Symptoms

Consistent with the extant literature, higher levels of parental stress were associated with being a single parent, pre-pandemic mental health problems, pre-pandemic substance use problems, strain from changes in the relationship with a partner and, most notably, lower levels of family functioning and higher levels of worries about finances. Moreover, strain from changes in relationships with friends, working from home, household workload and health worries—all pandemic-related stressors which had, to our knowledge, not been studied regarding their association with parental during-pandemic stress levels—were also associated with higher during-pandemic stress levels. Moreover, parents who reported higher strain from working from home also had higher levels of strain from household workload. Worries about finances were associated with health worries, possibly mirroring the fear that health problems could diminish one’s capacity to secure a sufficient income. Moreover, both worries may be influenced by levels of trait anxiety. Interestingly, actual income loss was not too strongly related to worries about finances (r = 0.151 **). A reason may be that health worries could be related to possible future income losses rather than current ones. Previous research into child and family well-being has highlighted the role of economic pressure. Economic pressure seemed to promote interparental conflict and parental stress and psychosocial symptoms [15]. Associations that fit this pattern can be observed in the present sample. Worries about finances were associated with strain from changes in relationships with partners. When strain from the relationship with a partner was high, those with children tended to be high as well (r = 0.419 **). Although caution needs to be exercised, because current data are cross-sectional and do not allow for causal or sequential interpretations, it seems like the pattern in how economic strain burdens family well-being can be retraced in the current data. Further research on the influence of economic pressure on family well-being with longitudinal data would be desirable to find out more about potential mechanisms both in and beyond the COVID-19 pandemic.

### 4.3. Findings on Help Seeking and Help Need

Participating families in the sample were separated into those reporting no during-pandemic help need (76.2% of answering participants), those who had used services for help prior to the pandemic and who reported a help need in the pandemic (continuous help need 11.3%), and those with no prior contact with help services but a during-pandemic help need (new help need; 12.5%). Across all these groups, participating parents reported a mean drop in mental and physical health between February and April 2020, but the drops were especially strong in families with a new help need. So, while the majority of families at this point during the pandemic reported no help need, nearly a quarter of families wished for help services in the first wave of the COVID-19 pandemic in the current sample. As assumed, there is a significant proportion of families among them who seem to struggle continuously, resulting in the families’ wish for professional help or counselling before and during the pandemic. However, about half of the families who reported a during-pandemic help need had not accessed help services before. We therefore argue that with the emergence of the COVID-19 pandemic the group of families reporting a help need has markedly expanded. Family help providers should be enabled to adjust their capacity to prepare for an increased help need among families.

### 4.4. Findings on Comparison of Different Help Need Groups

In order to characterise families with a during-pandemic help need further, we were especially interested in the interplay of known general family risk factors and pandemic-related factors with their rather unique challenges. Specifically, in order to meet the help need of newly struggling families, better knowledge of their characteristics could (a) help prepare family help providers and (b) help design interventions to support them adequately.

When comparing families with no reported help need, a continuous pre- and during-pandemic help need and a new during-pandemic help need we found several differences. Although previous studies found a link between housing and the presence of young children and COVID-19-related stress, which we also hypothesised, there was no difference across the three described groups. Yet, as hypothesised, there were differences with regard to single parenthood: proportions of single parents were highest among families with a continuous help need, followed by those with a new help need and lowest in families with no help need. A similar pattern could be obtained for during-pandemic mental health symptoms and family functioning: parental mental health symptoms and stress and family relationships were poorest among families with a continuous help need, followed by those with a new help need and most functional among families with no help need. The distribution of parents with a pre-pandemic diagnosed mental health problem is similar: more than half of families with a continuous help need report a pre-pandemic parental mental health diagnosis (56.25%), while the proportion was smaller in families with a new help need (30.19%) and smallest among families with no help need (13.00%). These findings are rather consistent with our hypotheses. We believe current findings are in line with a vulnerability-stress-model: families with a new help need had a vulnerability for reporting a help need due to heightened levels of pre-pandemic parental mental health problems. Now, with the added stress of the pandemic, they are showing symptoms of dysfunction resulting in a new reported help need.

Yet, substance use problems did not play a role in families with a new help need, both before the pandemic and during the pandemic. However, pre-pandemic substance use problems were prevalent in 14.58% of families with a continuous help need. This is in line with previous literature which underlined the association between parental substance use problems and family dysfunction.

Among the pandemic-related stressors reported by families with a new help need, we observed the biggest decrease in both mental and physical health, more worries about finances and reported heightened levels of stress through household workload. For families with a continuous help need, pandemic-related stressors which impacted families were household workload and health worries. Other factors which we thought might play a role across the different family groups did not, at least not significantly. Among these were worries about job security, working from home, strained partner relationships, strained relationships with friends and proportions of families who did not leave their house. Regarding strained partner relationships and relationships with friends, the overall test did not show significant differences, but relationships tended to be more strained in families with a during-pandemic help need. It remains noteworthy that the proportion of families not leaving the house at all was twice as high in the new help need group (11.3%) than in the no help need group (5.6%).

### 4.5. Findings on Prediction of during-Pandemic Help Need in Light of Previous Help Need

The above mentioned influences were largely confirmed in multinomial analyses, although the influence of household workload was not strong. For families with a new help need, changes in parent physical health were specifically important. For families with a continuous help need, health worries were particularly important. Both pandemic-related stressors remained important even when adding during-pandemic stress and pre-pandemic mental health problems in subsequent prediction models.

Moreover, the analyses underlined the importance of pre-pandemic mental health problems in parents with a during-pandemic help need. As stated above, parent mental health is a known general risk factor for family dysfunction. While many efforts are taken to design interventions specifically for families affected by parental mental health problems, in our eyes the importance of parent mental health stands out here yet again. At the same time, a during-pandemic help need could not solely be explained by pre-pandemic parental mental health problems.

What is true for both during-pandemic help need family groups is the role of during-pandemic stress. While during-pandemic stress significantly predicted help need, its relative impact remained very small in models with pre-pandemic parental mental health (continuous help need: OR = 1.18; new help need: OR = 1.07).

### 4.6. Limitations and Strengths

There are several limitations to the current study. Most notably, the results of the study are based on non-representative cross-sectional data. Comparisons of the current sample to representative German family studies were used to find out more about potential bias resulting from this. The present sample had a higher number of families with two or more children compared to single-child families [43]. Parent education levels were higher in the present sample, with 87.5% of families reporting high education, as compared to 44% in a representative reference study [51]. However, the current sample is comparable to reference data of other (pre-pandemic) representative family or parent samples with respect to child mental health and parents’ risky alcohol use. Oversampling rather well-off families is a phenomenon not new to family psychology studies. Interestingly, previous studies found, e.g., that parents with higher education levels seek help more readily because they have less fear of stigmatization, better knowledge of how to approach services and less scepticism regarding services. Assuming that the sample used here consists of rather better-off and less burdened families, a noteworthy proportion of families in the current sample still report a considerable burden. The need for help in the general population could be even greater.

Second, some of the data were asked for retrospectively, for example, accounts of pre-pandemic well-being, well-being in April 2020 or previous mental health diagnoses. We believe, however, that information asked for retrospectively was rather salient, such as a mental health diagnosis. Moreover, experiences in the earlier months of the pandemic may be well-remembered because of the exceptional quality of the pandemic’s beginning.

Third, some information was not obtained, such as a possible own COVID-19 infection or that of a close person. This may be specifically noteworthy, because a decrease in physical health between February and April 2020 turned out to be significant for families with a new help need. However, total infection cases with COVID-19 were relatively small at this early point in the pandemic in Germany. Up until the end of April 2020, 159.119 cases of COVID-19 had been registered [2], a fairly limited proportion of about 0.19% of the population. We therefore assume that the decrease in physical health seen in the current sample is not wholly attributable to COVID-19 infections.

Despite these limitations, the current sample is of considerable size and was gathered at a rather early time point during the COVID-19 pandemic. It gives insight into families’ well-being and adds a perspective on what factors generally put families under pressure, and what role pandemic-related experiences play. That is, we did not only measure stress levels or levels of well-being, but also included perceived sources of stress, some of which can be classified as specific to the requirements of the pandemic, such as health worries or increased household workload or working from home. Other factors we included in the present study are known to increase families’ vulnerability. This combined approach adds to a multi-faceted perspective on family well-being in times of challenge.

## 5. Conclusions

Given the study’s findings, we believe the need for family support services has increased during the COVID-19 pandemic. The COVID-19 pandemic is putting parents under heightened stress. As expected, at the time of the pandemic in question, levels of stress and mental health symptoms were highest among families with a continuous help need, but still high in families with a new help need. The association between stress levels and help need supports the notion that in the current sample, families with increased stress levels do perceive strain and look out for help. At the same time, it remains unclear how many families there are who are heavily burdened but do not want any help. Family support services should be accessible to families, even under social distancing rules. Moreover, we believe that resources for services need to be expanded to meet demand. When offering help, service providers should take into account families’ general risk constellation, specifically, possible parent mental health problems. However, they should also recognise the added strain of the pandemic on families, independent of pre-pandemic parental mental health. This is especially true for strain felt from changes in the physical health of parents and health worries, but also strain from household workload and financial worries. Underlining the stressful nature of the pandemic might make it easier for families to seek help, because fear of stigma may be reduced. Results from the present study could also help agents developing new interventions to support families in pandemics to devise their ideas.

## Figures and Tables

**Table 1 ijerph-19-14159-t001:** Sample description and comparison to pre-pandemic representative family samples in Germany.

	Sample Description		Pre-Pandemic Reference Data (German Representative Samples)	Sources
		95% CI		
Parent gender, *n* (%) (N = 602)				
Male	89 (14.8)			
Female	513 (85.2)			
Diverse	0 (0.0)			
Parent age, M (SD) (N = 598)	43.3 (6.56)			
Self-reported child gender (%) (*n* = 101)				
Male	51 (50.5)			
Female	49 (48.5)			
Diverse	1 (1.0)			
Self-reported child age, M (SD)	13.3 (2.15)			
# of children in household, *n* (%) (N = 602)				[43]
1	195 (32.4)	28.7–36.5	45.1% *	
2	290 (48.2)	43.9–52.0	43.0% *	
3 or more	117 (19.4)	14.60–24.13	12.0% *	
Parent educational level, *n* (%) (N = 603)				[51]
Low (erster Schulabschluss)	15 (2.5)	1.3–2.5	20% *	
Middle (mittlerer Schulabschluss)	60 (10.0)	7.6–12.3	24% *	
High (Fachhochschulreife or higher)	528 (87.5)	72.9–100.0	44% *	
During-pandemic stress (PSS-10) M (SD), 95% CI (N = 415)			M	[38]
Mothers	18.26 (7.11)	17.52–19.01	13.07 *	
Fathers	16.89 (6.56)	15.32–18.57	12.62 *	
Parent self-reported risky alcohol use (AUDIT-C; score ≥ 4 for females, score ≥ 5 for males; % (95% CI))				[43]
Mothers	43 (20.3)	16.2–24.6	18.4%	
Fathers	22 (31.4)	21.4–42.5	29.6%	
Child health-related quality of life (KIDSCREEN 10), M (SD), (*n* = 97)	46.70 (9.21)	44.90–48.51	49.85	[48]
Child psychosocial symptoms (SDQ), M (SD), (*n* = 91)			M	[49]
Emotional problems	2.97 (2.61)	2.46–3.55	2.73	
Conduct problems	1.48 (1.06)	1.26–1.71	1.74 *	
Hyperactivity/attention problems	3.57 (1.95)	3.18–3.96	3.46	
Peer problems	2.38 (1.90)	2.01–2.81	2.45	
Total problem score	10.41 (4.91)	9.46–11.43	10.37	

* *p* < 0.05.

**Table 2 ijerph-19-14159-t002:** Bivariate correlations of sociodemographic and variables on family health, stress, pandemic-related factors and family functioning.

	1.	2.	3.	4.	5.	6.	7.	8.	9.	10.	11.	12.	13.	14.	15.	16.
Measure																
1. During-pandemic stress (PSS-10)	1	0.729 **	0.163 **	0.103 *	−0.051	0.023	0.310 **	0.146 **	0.156 **	0.000	0.143 *	−0.423 **	0.301 **	0.161 **	0.300 **	0.276 **
2. During-pandemic mental health (SCL-K-9)		1	0.259 **	0.056	−0.070	−0.032	0.363 **	0.171 **	0.182 **	0.064	0.178 **	−0.437 **	0.361 **	0.077	0.271 **	0.269 **
3. Single-parent			1	−0.184 **	0.087 *	−0.031	0.216 **	0.095 *	0.246 **	0.063	0.065	−0.201 **	0.190 **	−0.001	−0.028	0.123 *
4. # of children under age 6				1	−0.152 **	−0.036	0.059	0.000	0.019	0.061	0.010	−0.069	0.059	0.090 *	0.124 **	−0.010
5. Housing (sqm/person)					1	−0.018	−0.052	−0.079	0.003	0.016	0.067	0.047	−0.020	0.056	−0.046	0.015
6. % income loss						1	−0.033	−0.0.006	−0.037	−0.070	0.023	0.045	0.151 **	−0.017	−0.029	−0.038
7. Pre-pandemic mental health problem							1	0.182 **	0.114 *	0.021	0.082	−0.033	0.087	−0.006	0.075	0.082
8. Pre-pandemic substance use problem								1	0.062	0.049	−0.026	−0.136 **	0.041	−0.100 *	0.056	0.119 *
9. Strained partner relationship									1	0.419 **	0.296 **	−0.159 **	0.157 **	0.028	0.174 **	0.054
10. Strained child relationship										1	0.272 **	0.000	0.060	0.047	0.130 **	0.007
11. Strained relationship to friends											1	−0.084	0.114 *	0.075	0.127 **	0.172 **
12. Family functioning (Family APGAR)												1	−0.121 *	0.045	−0.180 **	−0.149 **
13. Worries about finances													1	0.040	0.065	0.228 **
14. Working from home														1	0.259 **	−0.006
15. Household workload															1	0.143 **
16. Health worries																1

* *p* < 0.05, ** *p* < 0.01.

**Table 3 ijerph-19-14159-t003:** Family and parent characteristics of families with no help need, continuous help need and new help need.

	No Help Need (*n* = 323)		Continuous Help Need (*n* = 48)		New Help Need (*n* = 53)		ANOVA/Kruskal-Wallis
		95% CI		95% CI		95% CI	*p*
Sociodemographics							
Single-parent families	15.2 ^A,b^	(11.26–19.08)	47.9 ^b^	(33.78–62.05)	26.4 ^A^	(14.55–38.28)	0.000
# of children under age 6	0.41	(0.34–0.49)	0.38	(0.23–0.55)	0.49	(0.31–0.67)	0.660
Housing (sqm/person)	37.28	(35.54–39.04)	32.87	(29.58–36.11)	37.96	(32.24–45.16)	0.225
Pre-pandemic mental health							
Diagnosed mental health problem (%)	13.00 ^a,b^	(9.34–16.67)	56.25 ^b,c^	(42.22–70.28)	30.19 ^a,c^	(17.83–42.55)	0.000
Diagnosed substance use problems (%)	0.6 ^a^	(0.00–1.47)	14.58 ^a^	(4.60–24.57)	0.00	(0.00–0.00)	0.000
During-pandemic mental health symptoms							
Parental mental health (SCL-K-9)	5.59 ^a,b^	(5.02–6.18)	15.95 ^b,c^	(13.62–18.69)	10.33 ^a,c^	(8.11–12.76)	0.000
Stress (PSS-10)	15.71 ^a,b^	(14.89–16.50)	24.29 ^b,c^	(22.36–26.29)	21.00 ^a,c^	(18.82–23.27)	0.000
During pandemic alcohol use (AUDIT-C)	2.28	(2.04–2.53)	2.44	(1.78–3.17)	2.03	(1.42–2.70)	0.638
Pandemic-related stressors							
Worries about finances	3.28 ^a^	(3.20–3.37)	3.50	(3.27–3.74)	3.65 ^a^	(3.36–3.92)	0.004
Worries about job security	3.25	(3.16–3.35)	3.45	(3.15–3.72)	3.50	(3.21–3.82)	0.089
Working from home	3.74	(3.62–3.86)	3.93	(3.64–4.22)	4.00	(3.70–4.27)	0.191
Household workload	3.81 ^A,B^	(3.70–3.91)	4.16 ^B^	(3.85–4.43)	4.20 ^A^	(3.95–4.44)	0.004
Health worries	3.88 ^a^	(3.80–3.97)	4.43 ^a,b^	(4.21–4.65)	4.02 ^b^	(3.79–4.26)	0.000
Not leaving the house at all (%)	5.6	(3.08–8.10)	6.3	(0.00–13.37)	11.3	(2.79–19.85)	0.288
Loss in income in %	17.04	(5.50–33.40)	5.42	(2.35–8.83)	9.81	(5.65–14.11)	0.782
Change in parent mental health (February–April)	−0.56 ^a^	(−0.67–−0.45)	−0.73	(−1.07–−0.38)	−1.09 ^a^	(−1.39–−0.80)	0.001
Change in parent physical health (February–April)	−0.24 ^a^	(−0.30–−0.17)	−0.27 ^b^	(−0.50–−0.06)	−0.83 ^a,b^	(−1.08–−0.61)	0.000
Strained partner relationship	2.16	(1.99–2.33)	2.89	(2.33-3.49)	2.70	(2.13–3.25)	0.142
Strained child relationship	2.02	(1.85–2.21)	3.36	(2.74–4.00)	3.00	(2.44–3.63)	0.036
Strained relationship to friends	2.97	(2.79–3.15)	3.39	(2.88–3.82)	3.26	(2.74–3.80)	0.173
Family relationships (Family APGAR)	13.65 ^a^	(13.43–13.87)	11.81 ^a,B^	(11.02–12.62)	12.67 ^B^	(11.83–13.36)	0.000

Letters mark pairs of significant group differences: capital letters *p* < 0.05, small letters *p* < 0.01.

**Table 4 ijerph-19-14159-t004:** Multinomial logistic regression for help need group (no during-pandemic help need is reference group).

**Model 1**						
	**Continuous Help Need**			**New Help Need**		
	**OR**	**95% CI**	** *p* **	**OR**	**95% CI**	** *p* **
Block 1: Pandemic-related stressors						
Change in parent physical health	1.098	0.651–1.851	0.725	**0.420**	**0.269–0.658**	**0.000**
Worries about finances	1.201	0.774–1.864	0.413	1.391	0.927–2.086	0.111
Household workload	1.337	0.889–2.009	0.163	1.376	0.930–2.034	0.110
Strained child relationship	1.129	0.954–1.337	0.157	1.142	0.962–1.356	0.129
Health worries	**2.761**	**1.643–4.640**	**0.000**	1.043	0.666–1.361	0.855
Block 2: During-pandemic stress (PSS-10)						
Block 3: Pre-pandemic mental health problem Diagnosed mental health problem						
	Statistic	df	*p*			
Omnibus χ^2^	57.739	10	0.000			
Maximum Likelihood Estimation	393.119					
Nagelkerke’s R^2^	0.184					
**Model 2**						
	**Continuous Help Need**			**New Help Need**		
	**OR**	**95% CI**	** *p* **	**OR**	**95% CI**	** *p* **
Block 1: Pandemic-related stressors						
Change in parent physical health	1.165	0.675–2.010	0.583	**0.546**	**0.326–0.913**	**0.021**
Worries about finances	0.717	0.433–1.188	0.197	1.293	0.816–2.047	0.274
Household workload	0.993	0.618–1.569	0.977	1.142	0.738–1.768	0.551
Strained child relationship	**1.247**	**1.013–1.535**	**0.037**	1.171	0.970–1.414	0.100
Health worries	**2.069**	**1.173–3.649**	**0.012**	0.911	0.560–1.483	0.708
Block 2: During-pandemic stress (PSS-10)	**1.229**	**1.144–1.320**	**0.000**	**1.092**	**1.027–1.160**	**0.005**
Block 3: Pre-pandemic mental health problem Diagnosed mental health problem						
	Statistic	df	*p*			
Omnibus χ^2^	94.811	12	0.000			
Maximum Likelihood Estimation	385.834					
Nagelkerke’s R^2^	0.324					
**Model 3**						
	**Continuous Help Need**			**New Help Need**		
	**OR**	**95% CI**	** *p* **	**OR**	**95% CI**	** *p* **
Block 1: Pandemic-related stressors						
Change in parent physical health	1.080	0.619–1.885	0.787	**0.508**	**0.301–0.859**	**0.011**
Worries about finances	0.642	0.378–1.090	0.101	1.235	0.780–1.957	0.367
Household workload	1.020	0.629–1.656	0.936	1.179	0.761–1.826	0.461
Strained child relationship	1.228	0.991–1.522	0.061	1.165	0.965–1.407	0.111
Health worries	**2.055**	**1.154–3.659**	**0.014**	0.915	0.563–1.486	0.719
Block 2: During-pandemic stress (PSS-10)	**1.188**	**1.102–1.280**	**0.000**	**1.070**	**1.004–1.141**	**0.037**
Block 3: Pre-pandemic mental health problem Diagnosed mental health problem	**4.796**	**2.052–11.209**	**0.000**	**2.912**	**1.239–6.845**	**0.014**
	Statistic	df	*p*			
Omnibus χ^2^	110.133	14	0.000			
Maximum Likelihood Estimation	370.513					
Nagelkerke’s R^2^	0.368					

Significant predicors bold.

## Data Availability

The data presented in this study are available on request from the corresponding author.

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
