# Peer review of "Help Needs among Parents and Families in Times of the COVID-19 Pandemic Lockdown in Germany"

_ijerph, 2022, doi:10.3390/ijerph192114159_

Round 1
Reviewer 1 Report
I want to congratulate the authors for their work with relevance to public and mental health.
Improvement issues:
- the keywords are too many, you must select the most relevant ones;
- change the questions into a single question and set several objectives;
- Material and methods to put the time used in average to fill in the instruments;
- the results and discussion have very long sentences. It makes reading difficult;
- the conclusions are very incomplete and not very robust, they will have to be improved.
Author Response
Dear Reviewer 1,
Thank you very much for your time and effort to read and review our manuscript. We have inserted the points you made in bold below and answer after each point. Please find our answers in Italics.
I want to congratulate the authors for their work with relevance to public and mental health.
Thank you very much for this encouraging feedback.
Improvement issues:
- the keywords are too many, you must select the most relevant ones;
We have reconsidered the keywords and limited ourselves to a smaller number of keywords.
- change the questions into a single question and set several objectives;
We have followed your request: there is only one research question left. Several objectives of the article get into more detail.
- Material and methods to put the time used in average to fill in the instruments;
We have no data available regarding the average time it took to complete the survey, but we have pilot-tested survey completion and added the following information in the “Study design and procedure” section: “In a pilot run, parents and children took about 40 minutes to complete the survey each.”
- the results and discussion have very long sentences. It makes reading difficult;
We have scanned through both parts and tried to break up long sentences. We hope this makes reading easier.
- the conclusions are very incomplete and not very robust, they will have to be improved.
We have rethought the conclusion section again and have made amendments. At the same time, other reviewers mentioned they had been rather happy with the conclusions. So we hope to have balanced needs for rethinking as you suggested and communicating some of the ideas we had in mind in the first version of the conclusions-section.

Reviewer 2 Report
Help needs among parents and families in times of the COVID-19 pandemic lockdown in Germany – a cross-sectional study
The paper is well written and, in some aspect innovative. The introduction of subject is extremely long, it should be more objective. It is well known that COVID-19 impacted family life, generating stress, internal conflicts and financial concerns.
The innovation of the paper, in my view, was the comparison of the stressors caused by COVID-19 in relation to the families reported help need, before and during the pandemic.
Considerations:
1. Very long introduction;
2. The sample of n = 602 parents and their children appears only in the analysis of the data;
3. The discussion does not address either the sequence of tables, nor all variables analyzed
Author Response
Dear Reviewer 2,
Thank you very much for your time and effort to read and review our manuscript. We have inserted the points you made below and answer after each point. Please find our answers in Italics.
The paper is well written and, in some aspect innovative. The introduction of subject is extremely long, it should be more objective. It is well known that COVID-19 impacted family life, generating stress, internal conflicts and financial concerns.
The innovation of the paper, in my view, was the comparison of the stressors caused by COVID-19 in relation to the families reported help need, before and during the pandemic.
Considerations:
- Very long introduction;
We have shortened the introductory section by about a quarter and hope it is now more objective.
- The sample of n = 602 parents and their children appears only in the analysis of the data;
We have inserted an additional sentence on sample size in the “Study design and procedure” section.
- The discussion does not address either the sequence of tables, nor all variables analyzed.
We have made amendments to the discussion (in parts also to the results) to parallelize the sequence of results and tables and their discussion. We have also added subheadings more stringently and parallel to the results’ subheadings into the discussion. We hope this boosts readability. When checking back on the discussion, we have added the discussion of results which had not been addressed before.

Reviewer 3 Report
The submitted work is well-presented and provides valuable findings, however, there are a few aspects that should be taken into consideration before publishing in the journal of IJERPH. My minor comments and suggestions are given below:
The title of the manuscript doesn’t sound connected. Consider changing it to a more precise and understandable one.
Abstract
Methods should contain the methodology of gathering data and analysis adopted rather than saying a sample was collected online during the first wave of the pandemic. Rewrite.
The conclusion statement is attractive to its readership.
Introduction
There are reported claims starting from the second paragraph without any literature evidence. For example, the authors stated that results from a large cross-sectional study of parents and children ……. Help needs during the pandemic. No literature evidence….??
Materials and methods
How did the authors decide the selected sample size was enough and how the reliability of the chosen sample was checked?
What made you proceed with the selected sample size?
Results
Please mention under the tables what * means? What was the desired level of significance?
A similar requirement is warranted for table 2.
Also, in table 3, what are alphabets a, b, c, and B meant for? Please mention it underneath.
Discussion and conclusions
The discussion and conclusion are well-described and may not require any additional alterations.
There are a few grammatical errors that need to be fixed.
Author Response
Dear Reviewer 3,
Thank you very much for your time and effort to read and review our manuscript. We have inserted the points you made below and answer after each point. Please find our answers in Italics.
The submitted work is well-presented and provides valuable findings, however, there are a few aspects that should be taken into consideration before publishing in the journal of IJERPH. My minor comments and suggestions are given below:
Thank you very much for your encouraging feedback.
The title of the manuscript doesn’t sound connected. Consider changing it to a more precise and understandable one.
We have eliminated the “- a cross-sectional study” parts of the title in order to make it sound more coherent. We have integrated the cross-sectional nature of the study into the “methods” section of the abstract, as you had also raised concerns about being more clear on methods and analytic strategy there.
Abstract
Methods should contain the methodology of gathering data and analysis adopted rather than saying a sample was collected online during the first wave of the pandemic. Rewrite.
We have rewritten the abstract part and adjusted the abstract as a whole again so that it is a maximum of 200 words. It now reads:
Abstract: Background: The COVID-19 pandemic was accompanied by multiple disruptions in everyday lives of families. Previous research has underlined the negative impact of the pandemic on stress among parents and identified factors related to heightened levels of stress. Yet, several potential stressors have not been taken into account. Also, little is known about how general and pandemic-related stressors impacted help-seeking intentions for personal or family problems. Methods: We recruited n = 602 parents and their children (n = 101) for a cross-sectional online survey on parent, child and family well-being, stress and help need after the first wave of COVID-19 infections in Germany. Data were analysed using multinomial regression analyses to predict family help need, taking into account pre-pandemic help-seeking. Results: Parents showed high levels of stress, which were associated with pre-pandemic mental health, family functioning, pandemic related worries about finances, household workload and health worries. While 76.2% of families reported no during-pandemic help need, 11.3% reported of a help need before and during the pandemic and 12.5% of families without prior help need reported a new help need during the pandemic. Conclusions: Results of the present study underline the need for help service providers to adapt their offers.
The conclusion statement is attractive to its readership.
Thank you. Another reviewer had voiced concerns about the conclusion statement. We have rewritten parts of it but still hope to have communicated the broader ideas which we had addressed in the previous version, that you liked.
Introduction
There are reported claims starting from the second paragraph without any literature evidence. For example, the authors stated that results from a large cross-sectional study of parents and children ……. Help needs during the pandemic. No literature evidence….??
Other reviewers had suggested, that all in all, the introduction is very long. We have shortened it and also checked, whether claims we make are well-substantiated. As for the passage you are referring to here “Drawing from a large cross-sectional study of parents and children, we…”, this passage refers to the current manuscript/ paper, which we wanted to introduce here. We have reworded the sentence to make it more readily understandable. It now reads: “Drawing from a large cross-sectional sample of parents and children during the first wave of the COVID-19 pandemic in Germany, the current study seeks to add insight into…”
Materials and methods
How did the authors decide the selected sample size was enough and how the reliability of the chosen sample was checked?
What made you proceed with the selected sample size?
We had several objectives in mind when devising the CovFam study. First, we wanted to study levels of stress and mental health problems and be able to compare them to general (pre-pandemic) family/ parent/ child data. Second, we wanted to study associations between general and pandemic-related stressors to stress and well-being. And third, we wanted to study expressed help need in light of previous help-seeking behaviour among families. As these objectives were rather diverse, and in many parts exploratory in nature (e. g. there is not much research on help need in families in pandemics; and we really did not know, how many families there would be, e. g. expressing a during-pandemic help need), we felt it did not make sense to make a power-calculation based on one specific outcome. So what we did is that we wanted to set up the online survey within a reasonable and still compact time, in order to allow the sample to gather . We felt taking this approach would also allow for retrospective answers to be given in a reasonably homogenous mode, so that possible memory effects in participants do not vary too greatly between first and last participants. So instead of aiming for a certain sample size, we decided to let the online survey run for 8 weeks to allow for a sample to build and enable desired snowball effects in sampling to unfold. Within these 8 weeks we tried to contact as many possible participants and disseminating agents as possible (see “Study design and procedure”).
The reliability of the current sample was important to us. That is why we have largely used measures, which had been used in representative samples of German parents, children and families before. This approach allows for comparisons between our current sample and (pre-pandemic) representative German samples. Our comparisons showed that the current sample in some areas (e. g. child mental health problems) were very comparable to previous data. In other variables (e. g. parental stress levels), we had anticipated effects of the pandemic, so they were elevated, but in a pattern that was very consistent to other during-pandemic research results. Of course, our comparisons also showed, that our (convenience) sample was different from representative German family data (see table 1), but this is true for many studies this early in the pandemic (e. g. Koopmann et al., 2020). We have discussed the comparisons and the sampling issue in the discussion as well.
Results
Please mention under the tables what * means? What was the desired level of significance?
We are sorry for this glitch in formatting. We have added the information underneath the tables as suggested. We have also added a statement about our chosen alpha-error level in the Analytic strategy section: “Significance was indicated by p < 0.05 for all statistical tests.”
A similar requirement is warranted for table 2.
We are sorry for this glitch in formatting. We have added the information underneath the tables as suggested.
Also, in table 3, what are alphabets a, b, c, and B meant for? Please mention it underneath.
We are sorry for this glitch in formatting. We have added the information underneath the tables as suggested.
Discussion and conclusions
The discussion and conclusion are well-described and may not require any additional alterations.
Thank you very much for your feedback. Two other reviewers suggested that we make some changes to the discussion, so there was a moderate rewrite there. We hope you are fine with this. We have also slightly amended the conclusions to satisfy another reviewer. We hope the broader and most important ideas are still communicated in the new version of the conclusions, We hope you are fine with this.
There are a few grammatical errors that need to be fixed.
We hope to have rectified all grammatical errors in the revised version of the manuscript.

Round 2
Reviewer 1 Report
The suggested changes were taken into account by the authors.
I consider for publication.
Reviewer 2 Report
I thank the authors for the rigorous revision of the manuscript. The manuscript was complemented with the necessary clarifications and will be a publication of great scientific relevance.